# Moral violations lead to demeaning: Non-disclosure of HIV undermines perceived psychological needs

Alireza Taqipanahi[1]*, Morteza Erfani Haromi[1], Fatemeh Shahri[2], Alexander Landry[3], Seyed Nima Orazani[4]

**1** Institute for Cognitive and Brain Sciences, Shahid Beheshti University, Tehran, Iran, **2** Department of Psychology, Shahid Beheshti University, Tehran, Iran, **3** Department of Organizational Behavior, Stanford University, Stanford, California, United States of America, **4** Department of Psychology, Carleton University, Ottawa, Canada

☉ These authors contributed equally to this work.
* alirezataqipanahi@gmail.com

## Abstract

Dehumanization of stigmatized groups is a pressing social challenge, and to effectively address it, we must understand how it arises. Here, we identify social-cognitive antecedents of a subtle form of dehumanization known as *demeaning*—which occurs when a target's "uniquely human" psychological needs (e.g., for meaning in life) are downplayed relative to their physiological needs shared with other animals. We study how demeaning arises by leveraging the Agent-Deed-Consequence (ADC) framework of moral cognition, which posits that perceptions of an Agent's Deeds, and the Consequences of these Deeds, independently shape perceptions of the Agent's moral character. Because morality is fundamental to perceptions of humanity, we reasoned that the perception of (im) moral character, in turn, would impact demeaning (i.e., downplaying the Agent's psychological needs). We support this notion in a vignette experiment in a context where stigma is rampant and crucially understudied—Iran. Participants (*N* = 272) evaluated a stigmatized Agent—an HIV-positive individual with a history of addiction. We varied the Agent's Deed (deceiving partner vs. being honest with a partner) and its Consequence (infecting partner with disease vs not) in a 2 x 2 design. Indeed, a negative Deed and Consequence led to greater perceived immorality. Immorality, in turn, influenced perceptions of the Agent's "uniquely human" needs, but not their "lower" physiological needs shared with animals. Moreover, our Iranian participants' perceptions of what is a "uniquely human" need differ from those in previous Western samples, underscoring the need for further investigation into the sociocultural forces influencing dehumanization.

**Data availability statement:** All data and code can be found at the following DOI: 10.6084/m9.figshare.27134031.

**Funding:** The author(s) received no specific funding for this work.

**Competing interests:** The authors have declared that no competing interests exist.

## Moral violations lead to demeaning: Non-disclosure of HIV undermines perceived psychological needs

Stigmatized groups—including drug addicts, homeless individuals, and criminals—are frequently dehumanized (i.e., perceived as lacking human attributes; [1–4]). While this phenomenon is well-documented, the psychological mechanisms driving it remain poorly understood. Given that morality is a cornerstone of human identity [5–7], and dehumanization is often associated with the attribution of immoral traits to stigmatized groups [8,9], moral perception emerges as a crucial factor in this process [10–12].

We examine this interplay in the context of HIV-positive individuals with addiction—a group subjected to layered stigma due to associations with "immoral" behaviors (e.g., drug use, unprotected sex) [13], particularly in non-Western contexts. Cross-cultural research highlights divergent stigma patterns: While Western societies have increasingly medicalized HIV (e.g., U = U campaigns) [14,15], in Iran, it remains heavily moralized, with HIV seen as a marker of deviance [16–18]. Limited public education, lack of legal protections, and the moralization of illness contribute to heightened stigma and social exclusion [16]. These cross-cultural differences in the moral vs. medical framing of HIV may affect how dehumanization is expressed. For instance, Iranian respondents may be more likely to interpret HIV status as evidence of immoral character, reinforcing dehumanization. To address this, our study examines how dehumanization occurs through a moral lens, using the Agent-Deed-Consequence (ADC) model from moral psychology. We propose a culturally-informed moral-psychological framework for understanding how dehumanization of HIV-positive drug addicts operates in Iran. Specifically, we assess how psychological and physiological needs are differentially attributed to a stigmatized target based on their actions and the consequences—revealing how demeaning, as a form of dehumanization, emerges in a moral context.

### Demeaning as a form of Dehumanization

Dehumanization—the denial of complex emotional and cognitive capacities to others—represents a core mechanism in stigma maintenance [7,19,20]. This phenomenon manifests across diverse marginalized groups: released prisoners (reduced sensitivity to social pain) [2], Black individuals (diminished perception of social pain) [21], people with disabilities (denied emotional depth) [22], and homeless populations (deprived of "uniquely human" attributes) [1]. Such dehumanization consistently predicts adverse outcomes, including reduced support and help-giving behaviors [1,2].

Individuals with substance use disorders experience particularly intense dehumanization compared to other stigmatized groups [3]. Research indicates they are perceived as more dangerous and morally culpable than persons with mental or physical disabilities, often eliciting stronger blame attributions [22,23]. This perception stems in part from lay theories suggesting that addiction fundamentally alters moral character, rendering individuals qualitatively different from their pre-addiction selves [24]. Empirical work by [25] demonstrates that addicted persons are systematically

attributed fewer essential human mental capacities relative to other marginalized populations, reflecting profound dehumanization [26]. Neuroscientific evidence further suggests this process may involve deactivation of the Theory of Mind network – neural circuitry crucial for social cognition – when observers evaluate individuals with addiction [25–27].

Dehumanization not only involves assessments of individuals' cognitive capacities but also extends to perceptions of their needs and motivations. [28] demonstrated this phenomenon, showing a tendency to downplay the importance of psychological needs in favor of physical needs when evaluating stigmatized groups, such as homeless people and addicts, a tendency they termed "demeaning." According to [28], the capacity to value psychological needs (e.g., meaning in life) is linked to a sophisticated, humanlike mind, while physical needs (e.g., food and shelter) can be attributed to any physical entity, including nonhuman animals. As a result, individuals perceived as having reduced mental capacities—such as drug addicts—are often subjected to demeaning treatment, where they are seen as motivated primarily by physical needs rather than the complex psychological needs associated with fully developed humans. In this way, viewing others as driven by only basic needs subtly strips them of human qualities, reinforcing a broader tendency to dehumanize stigmatized groups by perceiving their needs and motivations as lower, less sophisticated, and fundamentally different from those of typical humans.

The stigma of addiction sometimes intersects with that of HIV, as sexual relationships with addicted partners can transmit the virus. STIs (Sexually Transmitted Infections) like syphilis, gonorrhea, and HIV/AIDS have long evoked moral panic and hostility, resulting in the marginalization of STI-positive individuals [29,30]. This process occurs rapidly, with nonverbal cues alone sufficient to trigger dehumanizing responses toward STI-positive persons [31], consistent with the Prophylactic Dehumanization Model's prediction that disease-associated individuals are automatically processed as less human [32]. The confluence of addiction and HIV stigma creates a particularly pernicious form of compounded discrimination, wherein affected individuals face dual dehumanization based on both substance use and perceived contagion risk. Despite the clinical and social significance of this intersection, the unique dehumanization dynamics affecting HIV-positive individuals with comorbid addiction remain poorly characterized. The current study seeks to elucidate the psychological mechanisms driving dehumanization in this critically understudied population.

## Morality as an underlying mechanism

Despite extensive documentation of the dehumanization of stigmatized groups, we know little about what drives it (though there are important exceptions, e.g., [33]). Possessing moral values is often seen as a hallmark of being "uniquely human" [12], marking the transition from a self-centered, animalistic nature to a higher level of human sensibility. Indeed, an intrinsic moral compass is frequently regarded as essential to humanity [5,6] and features prominently in foundational models of dehumanization [11,12,32]. Indeed, people who violate core moral principles are often perceived as less than human (e.g., [5,29,34–36]).

The connection between perceived (im) morality and dehumanization is particularly relevant for stigmatized groups, such as HIV-positive individuals with addiction, where behaviors such as non-disclosure of serostatus are frequently construed as moral violations that compound existing stigma [37]. To systematically examine these dynamics, we employ the Agent-Deed-Consequence (ADC) model [38], an integrative theoretical framework that operationalizes moral judgment through three interconnected components: the character of the person (Agent), their actions (Deed), and the consequences of those actions (Consequence). These components are processed through heuristic cues drawn from three dominant ethical theories: virtue ethics, which focuses on the person's intentions and character; deontology, centered on moral duties and required actions; and consequentialism, which assesses the balance of harms and benefits resulting from the situation. By integrating these perspectives, the ADC model offers a structured yet flexible framework for understanding the multifaceted nature of moral evaluation. It aligns with dual-process theories in cognitive science [38], which suggest that moral judgments are often rapid, intuitive, and unconscious, with occasional conscious refinement [39,40]. The ADC model predicts that these components function together to shape moral judgments, with the weight of each

                                                                                   

component varying based on the context. Empirical validation of this framework comes from controlled studies examining scenarios of disease disclosure [39], which demonstrated that positive configurations across all three components (virtuous Agent, ethical Deed, beneficial Consequences) produced the most favorable moral judgments, while negative configurations yielded the strongest condemnations. Subsequent research has successfully applied this model to diverse behavioral contexts [40], confirming its robustness as an analytical tool.

Through application of the ADC moral model to HIV and addiction contexts, we posit that negative evaluations of deeds (e.g., non-disclosure) and consequences (e.g., disease transmission) erode moral judgments of targets, subsequently driving dehumanization. This process manifests specifically through the selective denial of complex psychological needs (e.g., meaning, autonomy) that distinguish humans from other species, while preserving recognition of basic physiological needs (e.g., sustenance, shelter) that are phylogenetically shared and morally invariant.

## The present research

This study investigates the psychological mechanisms underlying the demeaning of HIV-positive individuals with addiction through a moral-psychological lens. Our experimental approach applies the ADC (Agent-Deed-Consequence) model to examine how moral evaluations of actions (e.g., disclosure/non-disclosure) and health outcomes shape perceptions of need entitlement in this doubly stigmatized population. This inquiry addresses two critical gaps in the literature: (1) the paucity of research on moral drivers of dehumanization in stigmatized groups, and (2) the limited understanding of these processes in non-Western contexts, particularly Iran, where HIV transmission patterns and stigma dynamics present unique cultural specificities.

The Iranian context proves theoretically illuminating for several reasons. First, the moral framing of HIV in Iran—where infection is often viewed as divine punishment for "sinful" behaviors like drug use or extramarital sex [16–18]—creates distinct pathways for dehumanization compared to Western medicalized frameworks. Second, Iran's socioeconomic landscape, characterized by sanctions and challenges to healthcare access [41–43], may reconfigure traditional need hierarchies, potentially elevating security as a fundamental psychological need that modifies standard demeaning processes.

From a public health perspective, this research carries urgent implications. Iran's HIV epidemic, predominantly fueled by injection drug use and sexual transmission within close relationships [44], creates a vicious cycle where stigma-driven care avoidance [45] exacerbates transmission risks. Healthcare provider bias [17,18] and diminished social support [46,47] further compound these challenges, often rooted in demeaning perceptions that reduce patients to their disease status and addiction history. Our study posits that these practical consequences stem from fundamental moral-cognitive processes whereby negative evaluations of deeds and consequences systematically restrict attributions of human needs.

To explore these dynamics, we propose the following hypotheses:

1. Based on the ADC model's emphasis on deed and consequence as independent drivers of moral judgment, HIV-positive individuals with addiction will be judged as more moral when both their action (*deed*: disclosure) and outcome (*consequence*: health) are positive, compared to negative conditions.

2. Given that lower-level (i.e., shared with animals) physiological needs are considered basic and universally recognized, previous research suggests they are less influenced by moral evaluations [28]. Accordingly, we hypothesize that their attribution will remain stable across Deed and Consequence conditions.

3. Given that higher-level (i.e., uniquely human) psychological needs are uniquely human and strongly tied to moral evaluations [28], we expect their attribution to vary depending on both Deed and Consequence.

4. *Because morality is central to perceptions of humanity [6], we expect moral judgments to mediate the effect of Deed on higher-level psychological needs, with Consequences moderating this path.* This pattern is not expected for lower physiological needs (see Fig 1).

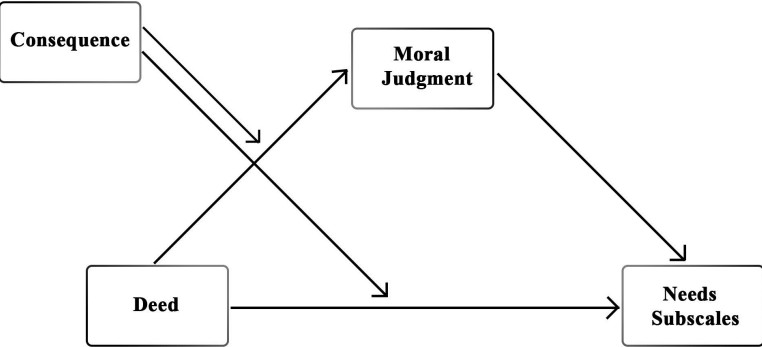

**Fig 1. Moderated mediation path diagram for the fourth hypothesis.**

## Materials and methods

### Participants

We estimated the power of the main hypothesis: moral judgment mediates the relationship between deed (lying or telling the truth) and perception of higher-level *psychological* needs, and the strength of this mediation effect depends on the consequence (the presence or absence of HIV). Based on [40], we hypothesized that deeds would exert a large-sized effect on moral judgment (*a*-path). To allow for sufficient power, we conservatively estimated that the effect of moral judgment on the perception of needs could be smaller (small-to-medium *b*-path). According to [48], the required sample size to detect an effect of a percentile bootstrap mediation with a power of.80 for these *a*- and *b*-path coefficients is $N = 123$. Three hundred eighty Iranian people visited the first survey page. Of these, 346 (93%) agreed to participate and completed the survey. To ensure the accuracy of the data, we removed 18 participants who responded inaccurately in the manipulation checks, assessing their understanding of the vignette. Of 328 individuals, 34 were excluded from the analysis due to incomplete responses on items below the 80% response rate threshold. Finally, according to the response time, 22 participants who responded too short or too long (an extreme outlier) were excluded (see Fig 2). The analytic sample comprises 272 Iranian respondents, of which 50.7% are female ($M_{Age} = 30$, $SD_{Age} = 7.66$, $M_{SES} = 4.78$, and $SD_{SES} = 1.59$).

### Procedure

This study was conducted online using the Qualtrics platform, with participant recruitment occurring between 15/08/2022 and 07/07/2023. Participation was voluntary, and respondents could withdraw at any time by closing their browser. They were informed that they would read a hypothetical vignette, after which they would be asked to evaluate person X morally and respond to a series of questions. Informed consent was implied by continuing past the initial information page, and participants were also made aware that they had the option to skip any question they did not wish to answer. It was not obtained because of the online and anonymous nature of written consent. After completing the consent form, they entered demographic information, and, in a 2 (positive deed vs. negative deed) x 2 (positive consequence vs. negative consequence) between-subjects design, we randomly assigned respondents to one of the four hypothetical vignettes. Each vignette varied along two dimensions (Deed and Consequence) and described an addicted person who was suspected of having HIV (Table 1). After reading the vignette, the manipulation check was presented, moral judgment was evaluated, and participants responded to the Needs Scale. Finally, participants were informed of the survey's purpose, and a debriefing was held (a schematic of the procedure is shown in Fig 2). The Australian National University Human Research Ethics Committee (ANU HREC) approved the study under protocol number 2022/075 on 05/08/2022. All data and code can be found at the following https://doi.org/10.6084/m9.figshare.27134031.

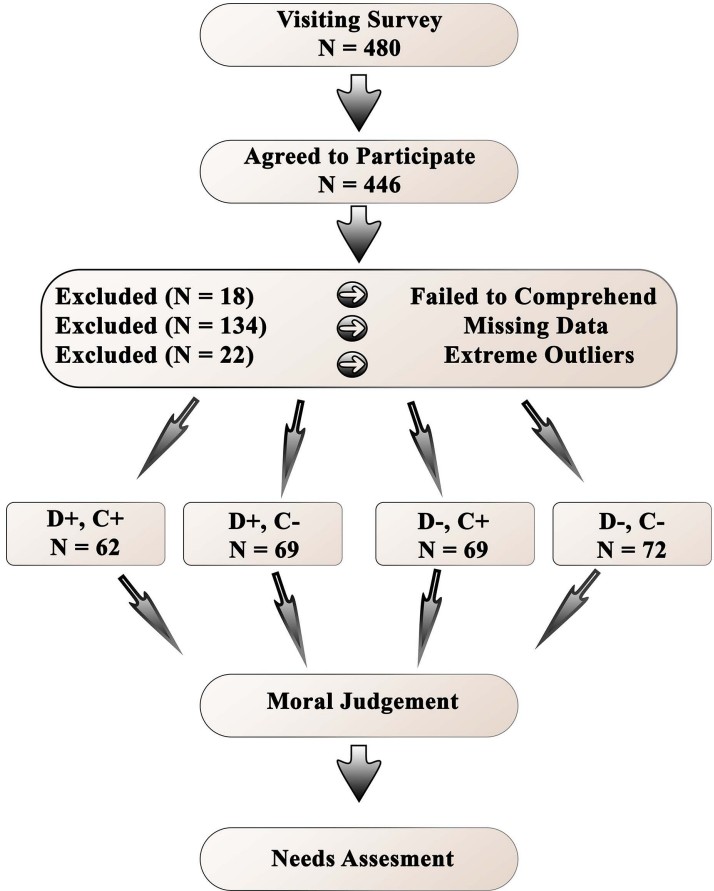

**Fig 2. The Flowchart of Data Screening and Data Collection.** Note. The procedure of data screening and data collection. D+ = positive deed; D- = negative deed; C+ = positive consequence; C- = negative consequence.

**Table 1. Vignette scenario with two dimensions and two levels each.**

Person X has been addicted for several years. He/she complains of weakness, lethargy, muscle aches, and night sweats when he/she goes to the health center. After the person declares that he/she is using shared bloody needles for injection, the doctor tells the person that he/she may have contracted HIV, a dangerous disease that has no cure and is transmitted through blood and sexual intercourse. To be sure, the doctor asked to have a blood test. After returning from the health center, he/she decides to [D⁻: *Lie* | D⁺: ***Tell the Truth***] to his/her wife about a diagnosis of his/her illness. Sometime later, the doctor informs him after observing the tests that [C⁻: ***he/she is sick and his/her wife also has the first symptoms of HIV*** | C⁺: ***he/she is healthy and the diagnosis of HIV was wrong***].

*Note.* Text in square brackets indicates the two experimentally varied vignette dimensions with negative and positive valence of D and C. In the survey, the text was neither bolded nor italicized.

## Materials

**Manipulation check.** A manipulation check, similar to the one used by [39], was employed to ensure that participants fully understood the vignette. The check consisted of three questions addressing key elements of the scenario: the agent's condition, actions, and consequences. Participants were asked to identify (1) the condition Person X was suffering from (options: HIV and addiction, syphilis, or flu), (2) whether Person X told the truth or lied about their illness, and (3) the outcome regarding the partner's health (whether the partner exhibited symptoms of HIV or both remained healthy).

Participants who answered any of these questions incorrectly were excluded from the final sample to maintain the validity of the responses and ensure comprehension of the vignette.

**Moral judgment.** Adopted from [39], after each vignette, we asked participants to rate the moral judgment of the situation by answering the following questions in random order: "How morally acceptable is what person X did in this situation, considering all the circumstances for yourself personally?" and "How morally acceptable is what person X did in this situation, considering all the circumstances for society?" The answer options ranged from 1, "*not at all,*" to 9, "*definitely*". Due to the very high correlation between both items for the HIV vignette ($r = .81$, $p < .001$), we calculated a mean value.

**Needs assessment.** *In this study, we conceptualized demeaning as a subtle form of dehumanization, in which individuals are perceived as lacking "uniquely human" psychological needs (e.g., meaning, self-actualization), while still being recognized as having basic physiological needs shared with other animals. To measure demeaning, we adopted a scale developed by Schroeder and Epley [28], consisting of thirteen items that assess the perceived importance of various needs for the target (Person X). The scale reorganizes Maslow's original five levels of needs into three broader categories based on the distinction between physiological and psychological needs. The lowest level includes physiological needs (e.g., "how important is the need to eat for Person X?"), followed by middle-level needs, which mix physiological and psychological aspects such as security (e.g., "how important is the need to have predictability in life for Person X?") and belonging (e.g., "how important is the need to feel loved for Person X?"). The highest level consists of purely psychological needs, including achievement and meaning (e.g., "how important is the need to achieve life goals for Person X?"). Items were presented in random order and rated on a 9-point Likert scale, with higher scores indicating greater perceived importance. To explore whether this three-level categorization holds true in the Iranian context or if any variations emerge, we conducted an exploratory factor analysis (EFA).*

**Sociodemographic information.** We assessed gender, age, and subjective social status to capture one's sense of one's place on the social ladder (measured using the MacArthur Scale of Subjective Social Status; 1 "lowest status," 10 "highest status"; [49]).

### Inclusivity in global research

Additional information regarding the ethical, cultural, and scientific considerations specific to inclusivity in global research is included in the Supporting Information (S1 Checklist).

### Results

Descriptive statistics of all dependent variables by condition were reported in Table 2. All continuous variables were z-standardized in linear regression models, and all numeric predictors were standardized and mean-centered in moderated mediation models. Since all interaction predictors are mean-centred, fixed effects represent the main effects. All

**Table 2. Descriptive Statistics of Dependent Variables by Condition.**

| Dependent Variables | NDNC | | NDPC | | PDNC | | PDPC | |
|---|---|---|---|---|---|---|---|---|
| | **M** | **SD** | **M** | **SD** | **M** | **SD** | **M** | **SD** |
| Moral Judgment (MJ) | 1.85 | 0.94 | 2.75 | 1.43 | 7.30 | 1.70 | 7.68 | 1.82 |
| Physiological Needs (PN) | 24.07 | 4.49 | 24.06 | 3.53 | 24.50 | 3.56 | 23.35 | 5.10 |
| Belonging (B) | 21.19 | 6.25 | 22.32 | 4.95 | 23.89 | 4.26 | 21.83 | 5.84 |
| Security – Actualization – Meaning (SAM) | 40.30 | 16.70 | 43.01 | 14.48 | 46.32 | 15.03 | 45.40 | 16.38 |

*Note.* M = Mean; SD = Standard Deviation; NDNC = Negative Deed, Negative Consequence; NDPC = Negative Deed, Positive Consequence; PDNC = Positive Deed, Negative Consequence; PDPC = Positive Deed, Positive Consequence.

analyses were done using the R programming language in RStudio. BruceR package in RStudio [50] using the *Interactions* package [51], while the *Mediation* package [52] was used for moderated mediation analyses. Analyses were considered significant at $p < 0.05$.

In a preliminary analysis, an exploratory factor analysis (EFA) using the principal axis factoring (PAF) method and Promax rotation was performed on the Needs scale. Descriptive statistics were reported in *Supplementary Table A1.* None of the items were removed based on their factor loadings. Three components had eigenvalues over Kaiser's criterion of 1 and, in combination, explained 67% of the variance (see the Scree plot *Supplementary Fig A1*). The items that cluster on the same components suggest that component one represents physiological needs, component two represents belonging needs, and component three stands for Security – Actualization – Meaning (SAM) needs (see *Supplementary Table A3* for factor loadings). In contrast to previous studies [28], which suggested that the needs scale might contain four subscales, our analysis revealed that all items were grouped into three components. Specifically, the security and actualization–meaning subscales were combined into a single cluster named Security–Actualization–Meaning (SAM). As a result, we organized the needs subscales based on these findings.

## Hypotheses 1–3: The effect of deed and consequence on moral judgment and the perception of needs

We conducted linear regression analyses to evaluate our hypotheses in the primary analysis. The predictors for these analyses included the Deed (D+, with D- as the reference), the Consequence (C+, with C- as the reference), and the interaction between these two factors. We also included the participants' gender, age, and SES as covariates. We considered four dependent variables in these analyses. The dependent variables were moral judgment (MJ) and personal Needs with three subscales, including Physiological Needs (PN), Belonging (B), and higher psychological needs of Security, Actualization, and Meaning (SAM). Moreover, pairwise comparisons (i.e., posthoc tests) with the Tukey method averaging over the levels of respondent's gender were used to find if the effect of one independent variable varies depending on the levels of the other.

To test our first hypothesis, we examined the effects of Deed (D) and Consequence (C) on moral judgment (MJ). As demonstrated in Fig 3, the model with Deed and Consequence as predictors and moral judgment as the outcome showed significant main effects of Deed, $\beta = 1.82$ [1.65, 1.98], $SE = 0.08$, $t(265) = 21.59$, $p < .001$, and Consequence, $\beta = 0.31$ [0.14, 0.47], $SE = 0.08$, $t(265) = 3.66$, $p < .001$. Thus, the target was perceived as more moral if they told the truth instead of lying (Deed) and if they and their spouse were healthy instead of sick (Consequence). However, the interaction effect of Deed and Consequence was not significant, $\beta = -0.19$ [-0.43, 0.05], $SE = 0.12$, $t(265) = -1.54$, $p = .126$, highlighting that when morally judging the target, the effect of Deed did not significantly change depending on the levels of Consequence or vice versa.

To test our second hypothesis, we examined the effects of Deed and Consequence on physiological needs. As shown in Fig 4, the model with Deed and Consequence as predictors and PN as the outcome revealed that neither the main effect of the Deed, $\beta = 0.12$ [-0.22, 0.45], $SE = 0.17$, $t(265) = 0.68$, $p = .497$, nor the main effect of the Consequence, $\beta = 0.01$ [-0.32, 0.35], $SE = 0.17$, $t(265) = 0.07$, $p = .943$, had significant effects on physiological needs. The interaction effect was not significant too, $\beta = -0.29$ [-0.77, 0.19], $SE = 0.25$, $t(265) = -1.19$, $p = .236$.

We examined two psychological needs models to test our third hypothesis: Belonging (B) and Security, Actualization, and Meaning (SAM). The B model (Fig 3) revealed that the main effect of Deed, $\beta = 0.49$ [0.16, 0.82], $SE = 0.17$, $t(265) = 2.96$, $p = .003$, was significant and positive, whereas the main effect of Consequence, $\beta = 0.20$ [-0.13, 0.52], $SE = 0.17$, $t(265) = 1.19$, $p = .237$, was not. The interaction of Deed and Consequence was also significant, $\beta = -0.59$ [-1.06, -0.12], $SE = 0.24$, $t(265) = -2.46$, $p = .014$, indicating that respondents considered the target to have lower belonging needs for negative than positive Deed when the Consequence of this Deed was negative, $\beta = -0.49$ [-0.82, -0.16], $SE = 0.17$, $t(265) = -2.96$, $p = .003$, but not when it was positive, $\beta = 0.10$ [-0.24, 0.44], $SE = 0.17$, $t(265) = 0.58$, $p = .566$. Moreover, when the Deed was negative, the attributed belonging was not significantly different in the case of negative and positive

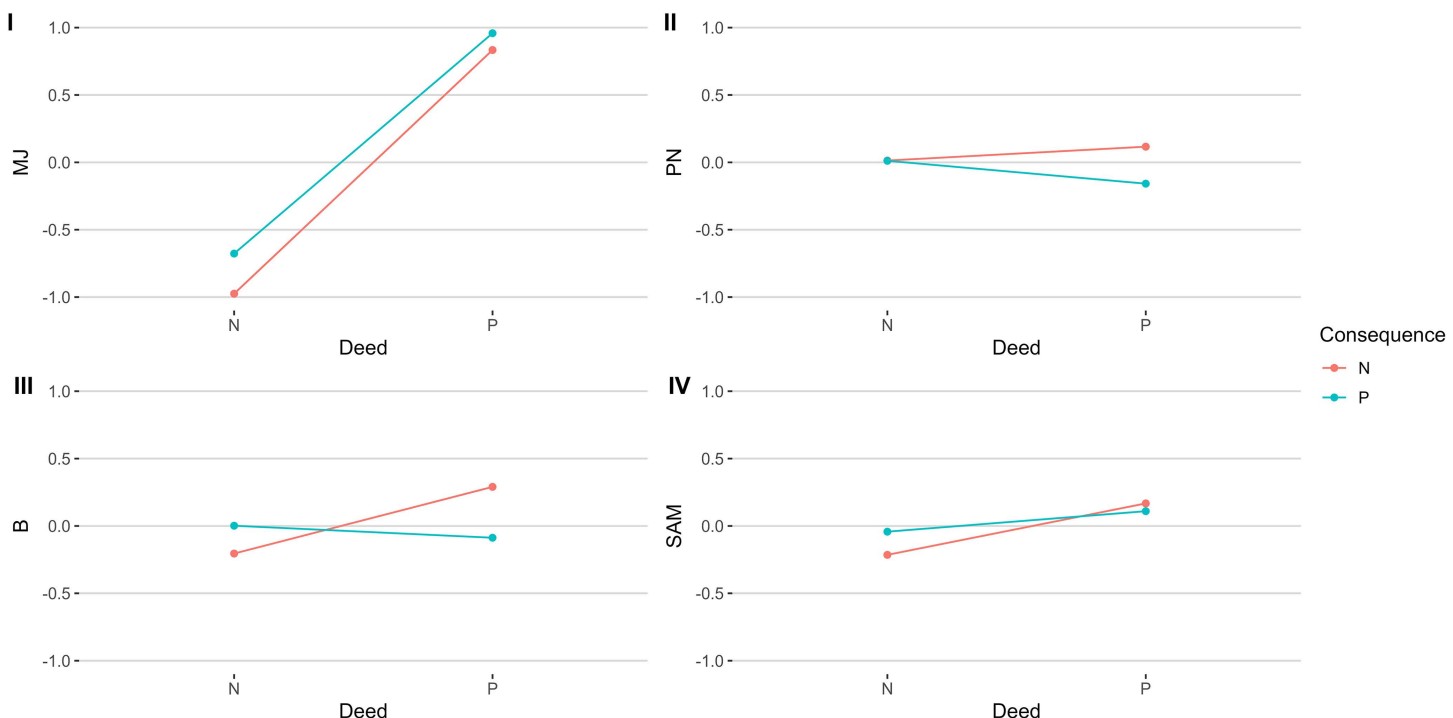

**Fig 3. Predicted Values for Dependent Variables Based on Different Conditions.** Note. Standardized predicted values for MJ (I), PN (II), B (III), and SAM (IV) depending on two levels (Positive, Negative) of the Deed and the Consequence. MJ = moral judgment; PN = physiological needs; B = belonging; SAM = security, achievement, and meaning.

Consequences, β = −0.20 [−0.52, 0.13], $SE$ = 0.17, $t$ (265) = −1.19, $p$ = .237. In contrast, when the Deed was positive, respondents considered higher belonging for the target in the case of negative compared to positive Consequences, β = 0.39 [0.05, 0.73], $SE$ = 0.17, $t$ (265) = 2.29, $p$ = .023. Similarly, in the SAM model (Fig 4), there was a main effect of Deed, β = 0.40 [0.07, 0.73], $SE$ = 0.17, $t$ (265) = 2.39, $p$ = .018. However, the main effect of Consequence, β = 0.19 [−0.14, 0.52], $SE$ = 0.17, $t$ (265) = 1.12, $p$ = .265, and the interaction effect, β = −0.27 [−0.74, 0.21], $SE$ = 0.24, $t$ (265) = −1.10, $p$ = .274, were not significant.

### Hypothesis 4: The effect of deed on psychological and physiological needs, meditated by moral judgment and moderated by consequence

To test our fourth hypothesis, moderated mediation analyses (PROCESS Model 8, [53]), including Deed (recoded: N = 0, P = 1) as an independent variable, moral judgment as the mediator, Consequence as the moderator, and gender, age, and SES as covariates, were performed with three dependent variables, namely physiological needs (PN), belonging (B), and SAM as needs subscales. Indirect effects, standard errors, and confidence intervals are estimated based on 5000 bootstrap samples, and percentile bootstrap confidence intervals were reported following moderated mediation analyses.

As depicted in Table 3, moral judgment was correlated with belonging and SAM (all $ps$ < .05) but not with physiological needs. As shown in Tables 4 and 5, analyses with physiological needs and belonging as dependent variables (Figs 4(I) and 4(II)) revealed that neither the conditional direct effects nor the conditional indirect effects of Deed on physiological needs or belonging were significant across both levels of Consequence when moral judgment was the mediator (all $ps$ > .05). When SAM was the dependent variable (Table 6, Fig 4(III)), the conditional direct effects of the Deed on SAM

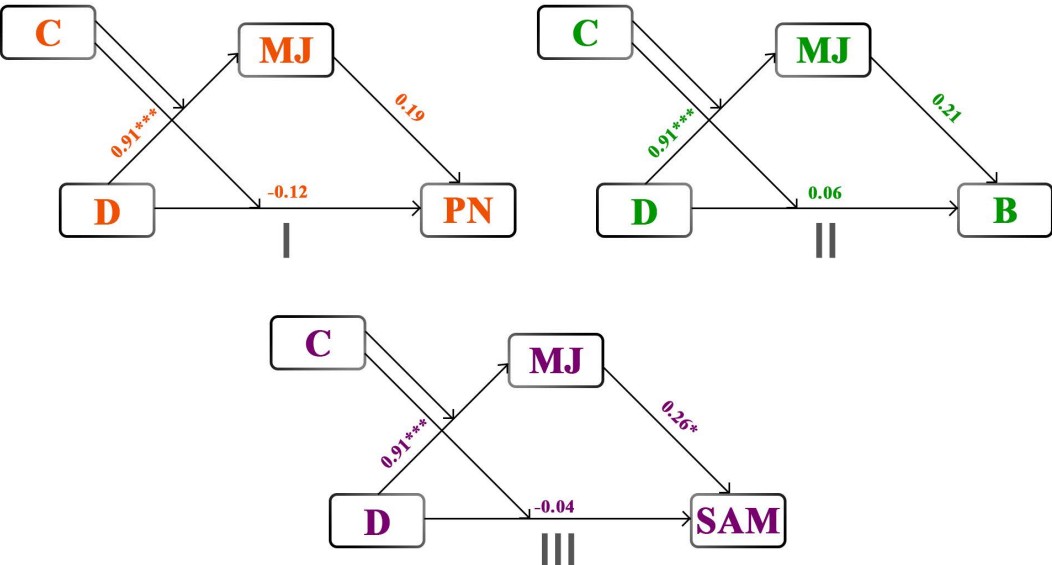

**Fig 4. Moderated Mediation Path Diagrams for Needs Subscales.** Note. Path diagrams describing moderated mediation models with PN(I), B(II), and SAM(III) as dependent variables. All path coefficients are standardized. MJ = moral judgment; PN = physiological needs; B = belonging; SAM = security, achievement, and meaning. * p < .05. ** p < .01. *** p < .001.

**Table 3. Descriptive Statistics and Correlation Matrix of Dependent Variables.**

| Variable | *M* | *SD* | 1 | 2 | 3 |
|---|---|---|---|---|---|
| 1. MJ | 4.79 | 3.02 | — | | |
| 2. PN | 24.01 | 4.20 | .03 | — | |
| 3. B | 22.31 | 5.44 | .14* | .35*** | — |
| 4. SAM | 43.68 | 15.76 | .19** | .33*** | .66*** |

*Note*. MJ = moral judgment; PN = physiological needs; B = safety, belonging; SEA = self-esteem, self-actualization.

* *p < 0.05.* ** *p < 0.01.* *** *p < 0.001.*

were not significant, whether the Consequence was negative, β = −0.04 [−0.31, 0.24], SE = 0.14, t (264) = −0.27, p = .785, or positive, β = −0.15 [−0.41, 0.11], SE = 0.13, t (264) = −1.11, p = .268. In contrast, the indirect effects via moral judgment were significant and positive in both negative, β = 0.24 [0.00, 0.44], SE = 0.11, z = 2.09, p = .037, and positive, β = 0.21 [0.00, 0.39], SE = 0.10, z = 2.10, p = .036, Consequence situations. Although moral judgment fully mediated the relationship between the Deed and SAM, these findings did not support the hypothesized moderated mediation effect in the SAM model as the Consequence did not moderate the mediation.

## Discussion

Although a variety of stigmatized groups are perceived as lacking complex mental and emotional capacities, little is known about the antecedents of such dehumanization. To address this gap, we adopted a moral lens to investigate how moral perceptions of HIV-positive individuals with a history of addiction influence demeaning—a form of dehumanization characterized by the tendency to downplay psychological needs in favor of physical needs. Using the ADC moral framework, we focused on how the Deed and Consequence shape moral judgments about the target and the attribution of physical and psychological needs. Furthermore, as dehumanization research has largely been restricted to Western settings,

**Table 4. PN-MJ Regression Models.**

|  | (1) PN | (2) MJ | (3) PN |
|---|---|---|---|
| Gender | −.044 | −.008 | −.045 |
|  | (.061) | (.030) | (.061) |
| Age | −.049 | −.036 | −.043 |
|  | (.061) | (.031) | (.062) |
| SES | .014 | −.008 | .010 |
|  | (.061) | (.030) | (.061) |
| Deed (D+, Ref. D-) | −.011 | .910*** | −.119 |
|  | (.061) | (.042) | (.141) |
| Consequence (D+, Ref. D-) |  | .109*** | −.085 |
|  |  | (.030) | (.062) |
| Deed: Consequence |  | −.065 | −.088 |
|  |  | (.042) | (.085) |
| MJ |  |  | .194 |
|  |  |  | (.124) |
| R2 | .005 | .758 | .023 |
| Adj. R2 | −.010 | .752 | −.002 |
| Num. obs. | 272 | 272 | 272 |

*Note*. Standardized regression coefficients are displayed, with standard errors in parentheses.

MJ = moral judgment; PN = physiological needs.

* $p < .05$. ** $p < .01$. *** $p < .001$.

**Table 5. B-MJ Regression models.**

|  | (1) B | (2) MJ | (3) B |
|---|---|---|---|
| Gender | −.007 | −.008 | −.009 |
|  | (.060) | (.030) | (.060) |
| Age | .100 | −.036 | .099 |
|  | (.060) | (.031) | (.060) |
| SES | −.120* | −.008 | −.127* |
|  | (.060) | (.030) | (.059) |
| Deed (D+, Ref. D-) | .104 | .910*** | .057 |
|  | (.060) | (.042) | (.137) |
| Consequence (D+, Ref. D-) |  | .109*** | −.066 |
|  |  | (.030) | (.061) |
| Deed: Consequence |  | −.065 | −.191* |
|  |  | (.042) | (.083) |
| MJ |  |  | .207 |
|  |  |  | (.121) |
| $R^2$ | .036 | .758 | .070 |
| Adj. $R^2$ | .022 | .752 | .045 |
| Num. obs. | 272 | 272 | 272 |

*Note*. Standardized regression coefficients are displayed, with standard errors in parentheses.

MJ = moral judgment; B = belonging.

* $p < .05$. ** $p < .01$. *** $p < .001$.

**Table 6. SAM-MJ Regression Models.**

| | (1) SAM | (2) MJ | (3) SAM |
|---|---|---|---|
| Gender | −.022 | −.008 | −.021 |
| | (.060) | (.030) | (.060) |
| Age | .003 | −.036 | .006 |
| | (.060) | (.031) | (.061) |
| SES | −.127* | −.008 | −.129* |
| | (.060) | (.030) | (.060) |
| Deed (D+, Ref. D-) | .136* | .910*** | −.038 |
| | (.060) | (.042) | (.138) |
| Consequence (D+, Ref. D-) | | .109*** | .001 |
| | | (.030) | (.061) |
| Deed: Consequence | | −.065 | −.075 |
| | | (.042) | (.084) |
| MJ | | | .262* |
| | | | (.121) |
| R2 | .035 | .758 | .057 |
| Adj. R2 | .021 | .752 | .032 |
| Num. obs. | 272 | 272 | 272 |

*Note*. Standardized regression coefficients are displayed, with standard errors in parentheses.

MJ = moral judgment; SAM = security, achievement, and meaning.

* $p < .05$. ** $p < .01$. *** $p < .001$.

examining this underexplored stigmatized group within Iran, a non-Western cultural context, offers valuable new insights into the cultural dynamics of dehumanization.

We specifically examined whether the three distinct levels of needs established by [28] in US culture hold true in the Iranian context or exhibit cultural variations. Our findings revealed that security, which [28] classified as a middle-level need, was closely linked with meaning and actualization—typically high-level and purely psychological needs in their study. This suggests that security functions as a high-level need in our sample and potentially more broadly within Iranian society. The heightened importance of security could stem from ongoing economic and political instability in Iran. Persistent inflation, high unemployment, and the long-term effects of international sanctions have devastated Iranian society and the economy, significantly raising tensions with the international community while constraining the country's economic growth [41,42]. Additionally, political instability has risen over the past three decades [54], further contributing to an environment of unpredictability. Economic and political factors undermine long-term planning and diminish many Iranians' sense of safety and predictability [43,55]. In this context, security emerges not just as a basic requirement but as an integral part of high-level psychological needs, closely interconnected with meaning and actualization—collectively referred to as SAM—while belonging remains a middle-level need and physiological needs are categorized as lower-level needs, consistent with [28] framework.

Morality is an evolutionary trait that distinguishes humans from other species [56,57]. Accordingly, comparing the evaluation of an HIV-positive individual's morality—based on their deed and consequence—with the attribution of three levels of needs provides critical insights. When patterns of need attribution align with moral judgments, they further reinforce the classification of these needs as high-level and uniquely human. Our findings revealed that moral judgments and the attribution of SAM needs followed similar patterns across various deeds and consequences, albeit with a slightly weaker

effect for SAM. This consistency underscores SAM's role as a high-level need. Conversely, the divergent patterns of moral judgment for physiological and belonging needs across conditions highlight their categorization as lower- or middle-level needs, reflecting their more fundamental and less morally contingent nature. As cultural understandings of "humanity" vary significantly across contexts [58], Our findings emphasize the significant role of cultural factors in shaping demeaning, a form of dehumanization. By applying moral psychology and dehumanization models to an underexplored, non-Western population, our study provides a critical foundation for future research to systematically explore how sociocultural dimensions influence dehumanization (see [59]).

Regarding the mechanism underlying demeaning, we posited that moral judgment might be the driving force behind dehumanization and would differentially influence how the three levels of needs—physiological, belonging, and SAM—are attributed to addicted HIV-positive individuals. Our findings suggest that perceived immorality, shaped by both deed and consequence, primarily affects higher needs—belonging and SAM—that are considered uniquely human rather than lower physiological needs, essential for basic bodily functioning, shared with animals. This distinction reflects a form of dehumanization known as demeaning, where individuals are viewed as driven primarily by basic physical needs rather than the higher psychological needs that define humanity [20]. Deed was found to play a significant role in attributing middle and high-level needs to the target, aligning with existing research that highlights the centrality of deeds in moral judgments [39,40]. However, our exploration of the mediating effect of moral evaluations (based on the Deed) on the three levels of needs revealed that moral evaluation did not mediate the attribution of physiological or belonging needs. This suggests that while moral judgments influence perceptions of character, they do not significantly shape assessments of these needs, irrespective of the consequences. In contrast, for SAM—classified as high-level needs in our study—moral judgment served as a significant mediator between Deed and SAM needs. Targets perceived as more moral based on their deed were attributed higher SAM needs, whereas immoral behavior resulted in lower attributions of these high-level needs. This underscores the critical role morality plays in distinguishing high-level, human-specific needs from lower- or middle-level needs, which are less morally contingent.

The fact that physiological needs remain unaffected by deed or consequence, while psychological needs such as belonging, safety, meaning, and actualization are influenced points to a clear distinction in how these needs are perceived. Recognizing physical needs involves simply acknowledging the presence of a body, whereas understanding psychological needs requires recognizing a human-like mind [28,60]. This disparity reflects a form of dehumanization—the belief that others are less driven by psychological needs, which are unique to humans, and more by basic physical needs shared with other animals [20]. Demeaning the importance of another's psychological needs aligns with treating them as having less uniquely human mental capacities [28]. Previous work shows that HIV-related stigma delays care-seeking [61], compromising health and increasing the risk of transmitting the virus to others [45]. Such delays are often linked to mistreatment by healthcare providers [16,17], which may stem from moral judgments about patients' actions and outcomes that foster demeaning perceptions. Underestimating the psychological needs of others can impair the ability to provide meaningful support [28], ultimately reducing social support and contributing to unequal treatment of HIV-positive individuals with addiction in Iran. These insights suggest that stigma-reduction efforts in Iran, where HIV and addiction are strongly moralized, must address moral attributions directly and reframe HIV as a health issue rather than a moral failing.

## Limitations and future directions

Our study encountered certain limitations. First, we kept the Agent, or "A" in the ADC model, constant and varied only Deeds and Consequences. Future research should consider varying the Agent's traits to understand better how moral judgments and need attributions are shaped by actions, consequences, and the nature of the individual involved. Additionally, while we examined a crucially understudied target group—HIV-positive individuals with a history of addiction—future work should expand this research to other stigmatized groups (e.g., homeless people, the poor, and criminals) to test

our findings' generalizability. Another limitation is that we employed a contrived vignette that lacked psychological realism. Future studies could benefit from using more immersive, realistic materials to capture participants' responses better. Finally, we relied on statistical mediation, which cannot definitively establish causal sequences. Future work could confirm our model through design-based alternatives to conventional mediation analysis (see [62]).

## Conclusion

The underlying mechanism of dehumanization can be understood through perceived morality. Our study demonstrates that dehumanization, specifically in the form of demeaning, operates through perceived immorality shaped by both deed and consequence. This primarily impacts higher-level needs—belonging, safety, actualization, and meaning—that are uniquely human, while lower physiological needs, essential for basic bodily functioning and shared with animals, remain unaffected. Importantly, the classification of security as a high-level need within the Iranian context highlights the significant role of sociocultural factors in shaping need attribution and dehumanization. These insights contribute to a broader understanding of how morality and culture intersect, shaping perceptions of humanity and providing a foundation for addressing dehumanization across diverse contexts.

## Supporting information

**S1 File. Check list.**
(DOCX)

## Author contributions

**Conceptualization:** Alireza Taqipanahi, Seyed Nima Orazani.

**Data curation:** Alireza Taqipanahi, Morteza Erfani Haromi.

**Formal analysis:** Alireza Taqipanahi, Morteza Erfani Haromi.

**Investigation:** Alireza Taqipanahi, Fatemeh Shahri.

**Methodology:** Alireza Taqipanahi, Seyed Nima Orazani.

**Project administration:** Alireza Taqipanahi, Fatemeh Shahri.

**Resources:** Alireza Taqipanahi.

**Software:** Alireza Taqipanahi, Morteza Erfani Haromi, Seyed Nima Orazani.

**Supervision:** Alireza Taqipanahi, Alexander Landry, Seyed Nima Orazani.

**Validation:** Alireza Taqipanahi, Morteza Erfani Haromi.

**Visualization:** Morteza Erfani Haromi.

**Writing – original draft:** Alireza Taqipanahi, Morteza Erfani Haromi, Fatemeh Shahri, Alexander Landry.

**Writing – review & editing:** Alireza Taqipanahi, Morteza Erfani Haromi, Fatemeh Shahri, Alexander Landry, Seyed Nima Orazani.

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
