## [Decision Letter · Decision Letter 0]

30 May 2025

Dear Dr. Taqipanahi,

As editor, I recognize the divergent recommendations but also the shared view that the manuscript addresses a potentially valuable topic. However, both reviewers agree that a more robust presentation of the theoretical framework and rationale, along with greater clarity in the methodological description and a more grounded interpretation of the findings, are needed.

Given the importance of the issues raised—particularly those highlighted by Reviewer 1—I am offering you the opportunity to revise and resubmit the manuscript under the category of major revision. This decision reflects my belief that the manuscript has potential, but only if you are able to substantially address the critiques provided.

I strongly encourage you to take into account all of the reviewers' comments—especially those raised by Reviewer 1—in your revision.

We look forward to receiving your revised manuscript.

Kind regards,

Marika Rullo

Academic Editor

PLOS ONE

Journal Requirements:

Reviewers' comments:

Reviewer's Responses to Questions

**Comments to the Author**

1. Is the manuscript technically sound, and do the data support the conclusions?

Reviewer #1: Partly

Reviewer #2: Yes

2. Has the statistical analysis been performed appropriately and rigorously?

Reviewer #1: Yes

Reviewer #2: Yes

3. Have the authors made all data underlying the findings in their manuscript fully available?

Reviewer #1: Yes

Reviewer #2: Yes

4. Is the manuscript presented in an intelligible fashion and written in standard English?

Reviewer #1: No

Reviewer #2: Yes

Reviewer #1: The present article addresses an important and under-researched topic: the perception of dehumanization toward HIV-positive individuals in Iran. While I recognize the value of extending research on such issues to understudied contexts beyond Western cultures—and acknowledge that this study has the potential to make a meaningful contribution, particularly given the scarcity of research on HIV-related stigma and disclosure in Iran—I must also point out that the manuscript, in its current form, is not yet ready for publication and requires extensive revision and rewriting.

The theoretical framing remains unclear: it is not evident whether the authors aim to develop new theoretical insights or to apply existing models to a specific cultural context. I strongly recommend that the authors clarify the applied nature of their research and restructure the manuscript accordingly, placing greater emphasis on the social relevance of HIV stigma and the specific challenges faced by HIV-positive individuals in Iran.

The methodological section also needs substantial improvement. It currently lacks clarity and coherence, with several key elements either underdeveloped or missing. For instance, the paper does not clearly explain how the authors conceptualize the specific form of dehumanization being investigated. If the focus is on dehumanization as demeaning, then the way this concept is operationalized and measured is not explicitly defined in the methods section.

Furthermore, the presentation of the hypotheses, methodology, and results would benefit from a clearer and more structured formulation. As it stands, the rationale behind each hypothesis is not always well-articulated, and their connection to the broader theoretical framework remains ambiguous.

Overall, while the topic is timely and relevant, and the study holds promise, the manuscript would greatly benefit from a more coherent structure, improved clarity of writing, and a more rigorous and transparent presentation of both the theoretical and methodological components.

Reviewer #2: The paper is generally well written and structured. However, in my opinion the paper has some shortcomings in regards to some sections.

In the introduction, it would be better to make a comparison between Western countries and Iran( as a none-western country) regarding the difference in the way they view the population living with AIDS.

in page 8, line 4,you talked about using Maslow's scale, but I didn't understand why you discussed it here and not in the method section.

the introduction is a bit lengthy. As such, I suggest the author reduces this section to keep only the most important elements.

This discussion could be expanded to explain more a bout cultural effects; in addition, discuss more a bout the implications of the study.

**Do you want your identity to be public for this peer review?** For information about this choice, including consent withdrawal, please see our Privacy Policy

Reviewer #1: No

Reviewer #2: No

---

## [Author Response · Author response to Decision Letter 1]

7 Aug 2025

Dear Dr. Marika Rullo,

Thank you for providing us with the opportunity to revise and resubmit our manuscript, Manuscript ID: PONE-D-25-03277, titled “Moral violations lead to demeaning: non-disclosure of HIV undermines perceived psychological needs.” Based on the reviewers’ helpful suggestions, we have revised the manuscript carefully. We have addressed all comments raised by the reviewers. Revisions in the manuscript are shown by tracker, and responses to each comment are listed below.

Reviewer #1:

The present article addresses an important and under-researched topic: the perception of dehumanization toward HIV-positive individuals in Iran. While I recognize the value of extending research on such issues to understudied contexts beyond Western cultures—and acknowledge that this study has the potential to make a meaningful contribution, particularly given the scarcity of research on HIV-related stigma and disclosure in Iran—I must also point out that the manuscript, in its current form, is not yet ready for publication and requires extensive revision and rewriting.

RESPONSE

We sincerely appreciate the reviewer’s thoughtful feedback and recognition of our study’s significance. We fully acknowledge the need for substantial revisions and have carefully revised the manuscript accordingly. We hope the changes reflect our strong commitment to improving the quality of the work.

1. The theoretical framing remains unclear: it is not evident whether the authors aim to develop new theoretical insights or to apply existing models to a specific cultural context. I strongly recommend that the authors clarify the applied nature of their research and restructure the manuscript accordingly, placing greater emphasis on the social relevance of HIV stigma and the specific challenges faced by HIV-positive individuals in Iran.

RESPONSE

Thank you for this valuable comment. Our study builds on prior work conceptualizing demeaning as a form of dehumanization and extends it by examining this process through a moral lens using the Agent–Deed–Consequence (ADC) model. This framework evaluates morality based on judgments about the actor, the action, and its consequences. We investigate how these moral evaluations influence the attribution of physiological versus psychological needs—a measure of demeaning—in HIV-positive individuals with addiction, focusing on the Iranian context where stigma is strongly moralized.

The introduction has been substantially revised to provide clearer theoretical framing. On page 3, we now briefly describe the specific challenges faced by HIV‑positive individuals in Iran and emphasize the social relevance of studying stigma in this context. In addition, the “Present Research” section explicitly explains how the ADC model is applied to examine demeaning, and the section has been streamlined to improve clarity and focus.

2. The methodological section also needs substantial improvement. It currently lacks clarity and coherence, with several key elements either underdeveloped or missing. For instance, the paper does not clearly explain how the authors conceptualize the specific form of dehumanization being investigated. If the focus is on dehumanization as demeaning, then the way this concept is operationalized and measured is not explicitly defined in the methods section.

RESPONSE

Thank you for your valuable feedback regarding the methodological section. We acknowledge the need for greater clarity in how we conceptualize and operationalize dehumanization as demeaning. In response, we have revised the “Needs Assessment” section (pages 12–13 of the manuscript) as follows:

“Needs assessment. In this study, we conceptualized demeaning as a subtle form of dehumanization, in which individuals are perceived as lacking “uniquely human” psychological needs (e.g., meaning, self-actualization), while still being recognized as having basic physiological needs shared with other animals. To measure demeaning, we adopted a scale developed by Schroeder and Epley [28], consisting of thirteen items assessing the perceived importance of various needs for the target (Person X). The scale reorganizes Maslow’s original five levels of needs into three broader categories, based on the distinction between physiological and psychological needs. The lowest level includes physiological needs (e.g., “how important is the need to eat for Person X?”), followed by middle-level needs, which blend physiological and psychological aspects such as security (e.g., “how important is the need to have predictability in life for Person X?”) and belonging (e.g., “how important is the need to feel loved for Person X?”). The highest level consists of purely psychological needs, including achievement and meaning (e.g., “how important is the need to achieve life goals for Person X?”). Items were presented in random order and rated on a 9-point Likert scale, with higher scores indicating greater perceived importance. To examine whether this three-level categorization holds in the Iranian context—or whether cultural variations emerge—we conducted an exploratory factor analysis (EFA).”

3. Furthermore, the presentation of the hypotheses, methodology, and results would benefit from a clearer and more structured formulation. As it stands, the rationale behind each hypothesis is not always well-articulated, and their connection to the broader theoretical framework remains ambiguous.

RESPONSE

Thank you for highlighting the need to present the hypotheses more clearly and link them to the theoretical framework. We have revised the “Present Research” section of the introduction (pages 8-9) to clarify the rationale behind each hypothesis. Specifically, we build on prior work conceptualizing demeaning as a form of dehumanization, where psychological (uniquely human) needs are downplayed relative to physiological ones (Schroeder & Epley, 2020). We integrate this with the Agent–Deed–Consequence (ADC) model of moral judgment (Dubijevic & Racine, 2014), which explains how evaluations of morality arise from judgments about the actor, the action, and its consequences. Combining these perspectives, we predict that moral evaluations shaped by deed and consequence will influence attributions of psychological needs (but not physiological needs). This logic guided the following hypotheses:

1. Based on the ADC model’s emphasis on deed and consequence as independent drivers of moral judgment, HIV-positive individuals with addiction will be judged as more moral when both their action (deed: disclosure) and outcome (consequence: health) are positive, compared to negative conditions.

2. Given that lower-level (i.e., shared with animals) physiological needs are considered basic and universally recognized, previous research suggests they are less influenced by moral evaluations [28]. Accordingly, we hypothesize that their attribution will remain stable across Deed and Consequence conditions.

3. Given that higher-level (i.e., uniquely human) psychological needs are uniquely human and strongly tied to moral evaluations [28], we expect their attribution to vary depending on both Deed and Consequence.

4. Because morality is central to perceptions of humanity [6], we expect moral judgments to mediate the effect of Deed on higher-level psychological needs, with Consequences moderating this path. This pattern is not expected for lower physiological needs.

4. Overall, while the topic is timely and relevant, and the study holds promise, the manuscript would greatly benefit from a more coherent structure, improved clarity of writing, and a more rigorous and transparent presentation of both the theoretical and methodological components.

RESPONSE

Thank you for this constructive feedback. We took this comment seriously and made major changes to the introduction. We reworked its structure to make it more coherent and rewrote parts of it so the ideas flow more naturally and the theoretical framing is clearer. Our goal was to make the writing easier to follow and convey the concepts more effectively.

Reviewer #2:

The paper is generally well written and structured. However, in my opinion the paper has some shortcomings in regards to some sections.

RESPONSE

We truly appreciate the reviewer’s positive assessment of the paper. We have carefully addressed the specific shortcomings noted in the comments and revised the relevant sections of the manuscript accordingly.

1. In the introduction, it would be better to make a comparison between Western countries and Iran (as a none-western country) regarding the difference in the way they view the population living with AIDS.

RESPONSE

We greatly value your insightful comment. We have revised the introduction to include a paragraph comparing perceptions of people living with HIV in Western contexts and Iran (page 3)

“Cross-cultural research highlights divergent stigma patterns: While Western societies have increasingly medicalized HIV (e.g., U=U campaigns) [14,15], in Iran, it remains heavily moralized, with HIV seen as a marker of deviance [16–18]. Limited public education, lack of legal protections, and the moralization of illness contribute to heightened stigma and social exclusion [16]. These cross-cultural differences in the moral vs. medical framing of HIV may affect how dehumanization is expressed. For instance, Iranian respondents may be more likely to interpret HIV status as evidence of immoral character, reinforcing dehumanization.”

2. in page 8, line 4, you talked about using Maslow's scale, but I didn't understand why you discussed it here and not in the method section.

RESPONSE

Thank you for your insightful feedback. In response, we removed the mention of Maslow’s theory from the introduction and incorporated it into the revised “Needs Assessment” section within the Method part (pages 12-13), where it more appropriately fits with the explanation of how demeaning was operationalized and measured.

3. the introduction is a bit lengthy. As such, I suggest the author reduces this section to keep only the most important elements.

RESPONSE

Thank you for your comment. We revised and shortened the introduction, removing around 500 words to keep only the most essential elements and improve readability.

4. This discussion could be expanded to explain more about cultural effects; in addition, discuss more about the implications of the study.

RESPONSE

We truly appreciate this helpful comment. To address it, we added the following paragraph at the end of the discussion:

“Previous work shows that HIV-related stigma delays care-seeking [61], compromising health and increasing the risk of transmitting the virus to others [45]. Such delays are often linked to mistreatment by healthcare providers [16,17], which may stem from moral judgments about patients’ actions and outcomes that foster demeaning perceptions. Underestimating the psychological needs of others can impair the ability to provide meaningful support [28], ultimately reducing social support and contributing to unequal treatment of HIV-positive individuals with addiction in Iran. These insights suggest that stigma-reduction efforts in Iran, where HIV and addiction are strongly moralized, must address moral attributions directly and reframe HIV as a health issue rather than a moral failing.”

---

## [Decision Letter · Decision Letter 1]

13 Oct 2025

Moral violations lead to demeaning: non-disclosure of HIV undermines perceived psychological needs

PONE-D-25-03277R1

Dear Dr. Taqipanahi,

We’re pleased to inform you that your manuscript has been judged scientifically suitable for publication and will be formally accepted for publication once it meets all outstanding technical requirements.

Kind regards,

Marika Rullo

Academic Editor

PLOS ONE

Additional Editor Comments (optional):

Reviewers' comments:

Reviewer's Responses to Questions

**Comments to the Author**

Reviewer #1: All comments have been addressed

Reviewer #2: All comments have been addressed

2. Is the manuscript technically sound, and do the data support the conclusions?

Reviewer #1: Yes

Reviewer #2: Yes

3. Has the statistical analysis been performed appropriately and rigorously?

Reviewer #1: Yes

Reviewer #2: N/A

4. Have the authors made all data underlying the findings in their manuscript fully available?

Reviewer #1: Yes

Reviewer #2: Yes

5. Is the manuscript presented in an intelligible fashion and written in standard English?

Reviewer #1: Yes

Reviewer #2: Yes

Reviewer #1: The authors have satisfactorily and thoughtfully addressed the reviewers’ comments, implementing a series of well-justified revisions that have substantially improved both the clarity and the overall quality of the manuscript. The responses provided demonstrate a careful engagement with the feedback, and I believe that the current version adequately and convincingly responds to the concerns raised in the previous round.

Reviewer #2: Dear authors,

Definitely, your revision met all the requested corrections and it was presented in an appropriate way.

**Do you want your identity to be public for this peer review?** For information about this choice, including consent withdrawal, please see our Privacy Policy

Reviewer #1: **Yes: ** Giovanni Telesca

Reviewer #2: No

---

## [Editor Report · Acceptance letter]

PONE-D-25-03277R1

PLOS ONE

Dear Dr. Taqipanahi,

I'm pleased to inform you that your manuscript has been deemed suitable for publication in PLOS ONE. Congratulations! Your manuscript is now being handed over to our production team.

Kind regards,

on behalf of

Dr. Marika Rullo

Academic Editor

PLOS ONE